# The 2-Minutes Walking Test Is Not Correlated with Aerobic Fitness Indices but with the 5-Times Sit-to-Stand Test Performance in Apparently Healthy Older Adults

**DOI:** 10.3390/geriatrics9020043

**Published:** 2024-04-01

**Authors:** Marina Gil-Calvo, José Antonio de Paz, Alba Herrero-Molleda, Arthur Zecchin, María Teresa Gómez-Alonso, Beatriz Alonso-Cortés, Daniel Boullosa

**Affiliations:** 1Faculty of Physical Activity and Sports Sciences, Universidad de León, 24007 León, Spain; magic@unileon.es (M.G.-C.); japazf@unileon.es (J.A.d.P.); arthurzecchin@gmail.com (A.Z.); mt.galonso@unileon.es (M.T.G.-A.); dboua@unileon.es (D.B.); 2Institute of Biomedicine, University of León, 24007 León, Spain; 3Faculty of Health Sciences, University of León, 24401 Ponferrada, Spain; balof@unileon.es

**Keywords:** aged, bicycle ergometry test, frailty, gait, muscle strength

## Abstract

The 2-minutes walking test (2-MWT) is a valid and reliable test that has a high correlation with the distance walked in the 6-minutes walking test (6-MWT). However, to date, no study has determined the relationship between 2-MWT performance and the aerobic fitness indices obtained during a maximal incremental test to confirm if this test is a valid surrogate of aerobic fitness in apparently healthy older adults. The main objective of this work was to identify the factors associated to the performance in the 2-MWT, including aerobic fitness, functional and spatial-temporal gait parameters. Seventeen elderly adults performed a maximal incremental cycling test to determine maximum oxygen consumption (VO_2max_) and ventilatory thresholds (VT1 and VT2), two static standing balance tests with open and close eyes, a 5-times sit-to-stand test (5-TSTS), a handgrip test, and a 2-MWT on three different days over 2 weeks. No correlations were found between aerobic fitness indices and the distance covered in 2-MWT, but significant moderate correlations were found between the distance covered in 2-MWT and the time to perform the 5-TSTS (rho = −0.49) and with stride length (rho = 0.52) during the test. In conclusion, the 2-MWT does not seem a good test to assess aerobic capacity while it showed to be associated to the 5-TSTS performance of the elderly.

## 1. Introduction

Walking tests are commonly used to assess individuals’ functionality, to monitor the effects of a treatment, and to determine the prognosis of an old individual [1]. There are different walking protocols, but the most studied and used is the 6-minute walking test (6-MWT) [1,2]. The 6-MWT is a valid and reliable tool for measuring gait function in older populations, which has exhibited moderate to high associations with various aerobic fitness parameters such as maximum oxygen consumption (VO_2max_) and maximum work capacity [1]. This association is expected as any continuous maximal effort lasting > 1 min mainly taxes the aerobic metabolism [3]. For these reasons, performance in the 6-MWT has been considered a valid surrogate of aerobic fitness, although maximal incremental tests are the gold standard for the measurement of this parameter. However, in some clinical settings, the 6-MWT is difficult to implement due to time and space commitments, therefore the 2-minute walking test (2-MWT) has emerged as a more suitable alternative with reference values available in the literature [2,4]. The 2-MWT has been also validated for older adults with frailty, providing a minimum detectable change of 7.7 m for this population [5]. Previous studies have confirmed that the distance completed in the 2-MWT presents a high correlation with the final distance completed in the 6-MWT [6,7].

To date, no relationship has been provided between any aerobic fitness parameter and performance in the 2-MWT in a sample of healthy older adults. Previously, one study [8] reported a weak correlation (r = 0.441) between the 2-MWT and VO_2max_ in a heterogeneous sample of multiple sclerosis patients with 23–68 years. More recently, a study [9] with a sample of older adults with cognitive frailty and other comorbidities found that the 2-MWT independently predicted VO_2max_ in both men and women. Therefore, we do not know if this expected relationship between aerobic fitness and performance in the 2-MWT can be confirmed in older adults without any clinical condition that may affect these relationships. Furthermore, there are no studies evaluating the relationship between submaximal aerobic indices (i.e., aerobic and anaerobic thresholds) and walking performance. This information would be important to confirm if performance in the 2-MWT is also valid to evaluate the aerobic fitness of healthy elderly individuals as the 6-MWT.

There are reference values for the 2-MWT as a test to identify impairments in walking ability, functional performance, and general function [2]. The 2-MWT is usually performed at the highest possible speed and gait speed is determined by two main factors, namely gait length and gait frequency [10]. In this regard, it has been suggested that the most important prerequisites for the ability to walk are the combination of strength and balance [8]. Several studies have found relationships between increased gait speed and greater performance in muscular strength, functional strength, and mobility [11,12,13]. In fact, it has been suggested that knee extensors strength correlates with step length and gait speed [11] and that ankle plantar flexors and hip extensors strength can predict gait speed [11,12]. Moreover, a recent study has also found a shorter stride length and an increased variability of spatial-temporal parameters in elderly people during the 2-MWT at their selected pace [14] which would lead to an increase in stride frequency as a compensation to increase walking speed [15], thus leading to an increased risk of falls [10,12,16]. Therefore, concurrent examination of spatial-temporal parameters during the 2-MWT and other strength and balance measures may serve to better identify the strengths and weaknesses of older adults in terms of gait and functionality, as well their potential associations.

Therefore, the first aim of the current study was to identify the functional and aerobic fitness factors associated to performance in the 2-MWT, to confirm if aerobic fitness indices are also related to its performance as reported in the 6-MWT. A secondary aim was to characterize walking gait to see if spatial-temporal parameters (i.e., double support, gait cycle, stride length, gait speed and gait frequency), would discriminate performance levels, and if these parameters may be related to strength and balance performances.

## 2. Materials and Methods

### 2.1. Subjects

A convenience sample of 17 older adults (12 women and five men) participating in a physical activity program for elderly people at the University of León, volunteered for participation in this study after receiving a formal invitation. Before evaluations, a physician conducted an anamnesis during 15–25 min to exclude those who presented any clinical acute or chronic condition precluding the completion of a maximal incremental test or limiting performance or safety in the other protocols of evaluation. These clinical conditions include severe cardiac pathologies, musculoskeletal pathologies that may limit the range of motion in the knee, hip, or lumbar spinae, and cognitive impairment identified by three or more errors in the Spanish version of Pfeiffer’s Short Portable Mental Status Questionnaire (SPMSQ) [17]. All the subjects were familiar with and had previously completed all the physical tests conducted in this study. After being informed of the procedures, all the subjects signed an informed written consent before participation.

The characteristics of the subjects are shown in Table 1, including blood testing parameters. The C-reactive protein was below 0.5 mg/dL in all cases except one subject.

### 2.2. Data Collection and Outcomes

The evaluations of the present study were conducted on three different days within a time span of 2 weeks and included extraction of blood samples after a minimum of 8 hours of fasting, body composition analysis, an incremental testing in cycloergometer until exhaustion, and the physical/functional evaluations which included static balance, handgrip dynamometry, sit-to-stand test and the 2-MWT.

Blood samples (20 mL) were sent to the laboratory (Eurofins, Megalab, Madrid, Spain) for analyses using standardized methods to determine a complete blood count (Sysmex XN-550, Sysmex, Norderstedt, Germany), glucose and lipid profiles (Vitros XT 3400 System Chemistry, Ortho, Copenhagen, Denmark), and inflammatory status (IL-6 Cobas 6000, Roche, Basel, Switzerland).

A stadiometer was used to measure height, a scale was used to obtain the body mass of the subjects, and the body mass index was calculated from these data. The percentage of body fat was obtained by means of a bone densitometry test (Prodigy Primo, General Electric^®^ and enCore 2009^®^ software version 13.20.033, USA).

Subjects performed a maximal incremental cycloergometer test, starting at 15 W and increased by 15 W every minute until volitional exhaustion or the inability to maintain a cadence of ~50 rpm, associated to a respiratory exchange ratio (RER) > 1.15 and a maximum heart rate (HR_max_) ≥ 85% of the estimated HR_max_ (220-age). A cycloergometer (Ergoline 900, Ergoline GmbH, Bitz, Germany) connected to a metabolic cart (Ergocard, Medisoft, Dinant, Belgium) was used. From the maximal test, the maximum power exerted (Max Power, W) by the subjects, the maximum oxygen consumption (VO_2max_, mL·kg^−1^·min^−1^), the oxygen consumption at ventilatory thresholds 1 (VT1, mL·kg^−1^·min^−1^) and 2 (VT2, mL·kg^−1^·min^−1^), and the power exerted at ventilatory thresholds 1 (pVT1, W) and 2 (pVT2, W) were obtained following the ventilatory equivalents of O_2_ and CO_2_ [18].

Subjects performed two 30 s static standing balance tests: a Romberg test with open eyes (ROE) looking at a target located 2 m away, and a Romberg test with closed eyes (RCE), on a force platform (Dinascan, IBV, Valencia, Spain) with a sampling rate of 1000 Hz. This is a valid and reliable test in this population [19]. Each test was performed three times. The average displacement velocity (m/s) and the swept area (mm^2^) of the center of pressure were determined. In addition, they performed two maximal repetitions of the handgrip test [20] (Jamar hydraulic hand dynamometer, Lafayette Instrument Company, Lafayette, IN, USA) standing with both arms alongside the body, holding the dynamometer with the dominant hand, with the palm of the hand facing the thigh and maintaining the application of maximum force (kgf) for 3 s, with at least 1 min of rest between repetitions. Finally, subjects performed two repetitions of the 5-times sit-to-stand (5-TSTS) test, with at least 2 minutes of rest between attempts. Subjects were asked to start sitting with their arms crossed over their chest and to stand up and sit down five times as fast as possible. The time (s) taken to perform the test was obtained by a trained evaluator with a manual stopwatch. The performance in this test can be considered a good surrogate of the lower limbs’ power following validated formulae [21]. The best attempt was selected for further analysis.

Finally, the subjects performed a 2-MWT on a 20-meter corridor with an optical system positioned at 7.5 and 12.5 m from the starting point to measure the spatial-temporal parameters of gait with a previously validated photoelectric system (OptoGait, MicroGait, Bolzano, Italy). Subjects were asked to walk as fast as possible during the test. Distance (m) completed by every subject was obtained as a measure of performance and the mean values of the frequency (strides/min), stride length (cm), stride time (s), average speed (m/s) and the percentage of the cycle in which the participant was in double support (%), were also obtained from the complete test [22,23].

### 2.3. Statistical Analysis

Normality data and homogeneity were assessed using the Shapiro-Wilk’s test. Spearman correlation coefficient (rho) with 95% confidence intervals (CIs), was used to determine the relationships between the performance in the 2-MWT and the outcomes from the other physical evaluations. An a priori alpha level was set at *p* < 0.05. The correlation coefficients were interpreted based on Pearson where rho < 0.10 indicated a negligible association, rho = 0.10 to 0.39 a small association, rho = 0.40 to 0.69 a moderate association, rho = 0.70 to 0.89 a high association, and rho ≥ 0.90 an almost perfect association. The analyzes were computed using the software IBM SPSS Statistics for Mac (Apple Inc., Cupertino, CA, USA), version 23.0 (IBM Corporation, Armonk, NY, USA).

## 3. Results

The descriptive data for each of the variables are shown in Table 2. When analyzing the correlations between the 2-MWT and the rest of the variables (see Table 2), a moderate negative association with the time to perform the 5-TSTS test and a moderate positive association with stride length and gait speed were found. No correlations were found with any of the outcomes related to maximum incremental cycloergometer test, static balance tests and handgrip dynamometry.

## 4. Discussion

The aim of this study was to identify the factors associated to the performance in the 2-MWT, trying to establish whether aerobic fitness indices, functional or spatial-temporal factors were related to the performance in this test. Previous studies found moderate to high relationships between the distance covered in the 6-MWT with the maximum work and the VO_2max_ determined in a cycloergometer test [1]. In our study, however, no correlations were found between the aerobic fitness indices obtained in the maximal incremental test and the distance covered in the 2-MWT in apparently healthy older adults. These results may imply that the 2-MWT is not a valid test for assessing aerobic capacity in elderly adults with no clinical conditions as evidenced in the anamnesis and the blood analyses. However, and contrary to our expectations, the performance in the 2-MWT exhibited moderate correlations with the stride length during the test and the 5-TSTS test performance, which can be explained due to the relationship between knee extensors strength and step length previously found [11], therefore suggesting that performance in this test is associated with lower limbs’ power in accordance with recent evidence showing a relationship among legs’ power, gait speed and frailty [24].

The 5-TSTS is a clinical test used as a valid measure of dynamic balance and lower limbs’ muscle power, in both healthy and clinical populations [25]. Of note, the power output in this test is expected to be considerably higher than the power associated to VO_2max_ during an incremental test in an ergometer, therefore these parameters represent different fitness components (i.e., anaerobic power vs. aerobic power). Importantly, it has been found that there is a moderate negative correlation between the distance covered in the 2-MWT and the time to perform the 5-TSTS, therefore suggesting that the 2-MWT would be more suitable as a neuromuscular test than an aerobic fitness test in this population. These results are in agreement with previous research in which the amount of force produced with a particular muscle (i.e., muscle strength) and the strength produced by multiple muscles used to meet the specific physiological demands of daily living activities (i.e., functional strength) of the lower limbs are associated with a high walking speed [11,12,13]. Meanwhile, no correlations were found with static balance tests, probably due to the lack of fall history in our sample, nor with handgrip strength which is a measure of general strength [20]. Thus, the distance completed in the 2-MWT has shown to be associated to the anaerobic power of the lower limbs in the elderly, but not with other functional and physical parameters. The validity of this test to detect changes in physical fitness components after an exercise intervention remains to be determined.

Interestingly, with respect to the gait spatial-temporal parameters, positive moderate correlations between the distance covered in the 2-MWT, stride length and gait speed were found in the current study. These results are in partial agreement with previous studies identifying cadence, step length and stride length as key determinants of gait speed during walking tests in older adults [26]. In our study, cadence does not seem to be related to distance covered in the 2-MWT. The fact that the correlations between gait speed, stride length and distance walked in the 2-MWT are moderate rather than high, and the lack of relationship with cadence, may be due to the fact that the spatial-temporal parameters were measured only in the central area (between 7.5 and 12.5 m) of the 20-m corridor, avoiding the areas where the subjects made turns, where the speed could be considerably reduced. Further research is necessary to confirm these results and to determine the optimal distance of the corridor for this test.

This study presents some limitations. First, the low n sample is an obvious limitation. However, all subjects are from the same physical activity program and represent a homogeneous sample of apparently healthy, physically active older adults with an excellent familiarization with procedures thus reducing the sources of bias. In addition, the aerobic evaluation was performed in the cycle ergometer which presents a different muscle activity and recruitment than an incremental walking test in the treadmill. All these aspects can be improved in further studies. Meanwhile, the current results provide useful information to better select the fitness evaluation tests for this population with consideration of the instruments available, its feasibility, validity and efficiency when examining specific fitness capacities in this population.

In conclusion, the current results suggest that the 2-MWT was not related to any aerobic fitness index in apparently healthy older adults. However, the performance in the 5-TSTS was associated with 2-MWT performance. These results confirm the relevance of anaerobic power for gait performance in the elderly while would suggest the necessity of using specific aerobic fitness testing for an appropriate evaluation of this capacity in this population.

## Figures and Tables

**Table 1 geriatrics-09-00043-t001:** Characteristics of the subjects.

Sample (N = 17)		
	Mean ± SD	95%CI
Age (years)	76 ± 6	(73–78)
Height (m)	1.55 ± 0.07	(1.52–1.58)
Body mass (kg)	61.3 ± 9.6	(57.5–65.1)
Body Fat (%)	33.1 ± 9.1	(29.5–36.7)
IMC (kg/m^2^)	25.9 ± 3.7	(24.4–27.4)
Blood sample results		
Hemoglobin (g/dL)	13.9 ± 1.3	(13.2–14.7)
Hematocrit (%)	43.4 ± 3.8	(41.4–45.5)
Cholesterol (mg/dL)	198 ± 21	(187–209)
HDL Cholesterol (mg/dL)	65 ± 18	(55–74)
LDL Cholesterol (mg/dL)	117 ± 21	(106–128)
Chol/HDL-Chol	3.30 ± 0.90	(2.82–3.77)
LDL-Chol/HDL-Chol	2.01 ± 0.82	(1.57–2.44)
Triglycerides (mg/dL)	84 ± 29	(69–100)
Glucose (mg/dL)	88 ± 19	(78–98)
HbA1c (%)	5.8 ± 0.9	(5.4–6.3)
HbA1c (mMol/mol)	40 ± 10	(35–46)
IL-6 (pg/mL)	9.95 ± 4.31	(7.65–12.24)

HDL: High-density lipoproteins; LDL: Low-density lipoproteins; HbA1c: Glycosylated hemoglobin; IL-6: Interleukin 6.

**Table 2 geriatrics-09-00043-t002:** Descriptive data of the different variables and correlations with distance covered in the 2-MWT.

	Mean ± SD (95%CI)	Min	Max	Median	Interquartile Range		Correlation with 2-MWT (rho; *p*-Value)
					25th	75th	
2 min walking test (m)	189.1 ± 21.6 (178.1–200.3)	142.5	215.0	195.0	173.5	205	-
5-times sit to stand test (s)	8.1 ± 2.2 (7.0–9.3)	5.6	16.0	7.7	7.1	8.4	−0.488; *p* = 0.047 *
Handgrip (kgf)	24.4 ± 10.1 (19.2–29.6)	8.8	52.0	22.0	19.0	28.0	0.11; *p* = 0.65
Incremental Test							
Max power (W)	90 ± 21.2 (79.1–100.9)	60.0	135.0	90.0	75.0	105.0	0.19; *p* = 0.45
VO_2max_ (mL·kg^−1^·min^−1^)	19.2 ± 3.7 (17.3–21.1)	11.8	27.7	18.6	17.1	20.9	0.16; *p* = 0.54
VT1 (mL·kg^−1^·min^−1^)	13.4 ± 3.7 (11.4–15.3)	6.0	22.0	13.0	10.5	15.5	0.20; *p* = 0.44
pVT1 (W)	51.2 ± 21.3 (40.3–62.1)	15.0	105.0	45.0	37.5	60.0	0.21; *p* = 0.41
VT2 (mL·kg^−1^·min^−1^)	17.8 ± 3.3 (16.1–19.5)	11.0	24.0	17.0	16.0	20.0	0.14; *p* = 0.58
pVT2 (W)	82.9 ± 22.6 (71.3–94.6)	30.0	135.0	90.0	67.5	90.0	0.37; *p* = 0.14
Spatial-temporal Gait Parameters							
Double Support (%)	23.5 ± 2.9 (22.0–25.0)	17.6	27.3	23.3	21.3	26.1	−0.04; *p* = 0.86
Gait Cycle (s)	0.82 ± 0.05 (0.79–0.84)	0.73	0.89	0.82	0.78	0.85	−0.05; *p* = 0.84
Stride Length (cm)	144.6 ± 15.1 (136.8–152.4)	118.3	170.7	141.0	133.8	158.9	0.52; *p* = 0.03 *
Gait Speed (m/s)	1.77 ± 0.20 (1.66–1.87)	1.33	1.96	1.87	1.63	1.93	0.53; *p* = 0.03 *
Gait Frequency (strides/s)	147.3 ± 9.1 (142.6–151.9)	135.2	164.7	146.0	140.5	153.6	−0.05; *p* = 0.84
Static Balance Test							
COP Swept Area ROE (mm^2^)	29.0 ± 14.4 (21.6–36.4)	10.0	56.1	25.9	17.4	43	−0.06; *p* = 0.81
COP Swept Area RCE (mm^2^)	35.1 ± 25.3 (22.1–48.1)	11.1	101.1	28.0	16.7	46.1	−0.30; *p* = 0.23
COP Mean Speed ROE (m/s)	0.01 ± 0.00 (0.010–0.013)	0.004	0.02	0.01	0.010	0.014	−0.05; *p* = 0.85
COP Mean Speed RCE (m/s)	0.02 ± 0.01 (0.013–0.020)	0.01	0.03	0.01	0.014	0.020	−0.18; *p* = 0.47

SD: Standard deviation; CI: Confidence interval; Max: Maximum; Min: Minimum; 2-MWT: 2 min walking test; VO_2max_: Maximum oxygen consumption; VT: Ventilatory threshold; pVT: power at ventilatory threshold; COP: Center of pressure; ROE: Romberg open eyes; RCE: Romberg closed eyes; * Statistical significance.

## Data Availability

The datasets generated during and/or analyzed during the current study are available from the corresponding author on reasonable request.

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
