# Peer review of "The 2-Minutes Walking Test Is Not Correlated with Aerobic Fitness Indices but with the 5-Times Sit-to-Stand Test Performance in Apparently Healthy Older Adults"

_geriatrics, 2024, doi:10.3390/geriatrics9020043_

Round 1

Reviewer 1 Report

Comments and Suggestions for Authors

This is a study assessing the (multiple aspects of the) validity of the 2MWT (e.g., construct, criterion, divergent, etc). This is an interesting study; however, I feel that the rationale for conducting it (Introduction) is relatively poor. Although authors present some arguments for testing the measurement properties of the 2MWT (e.g., correlation with 6MWT and others), the conceptualization of the 2MWT as a test/measure of aerobic capacity/fitness is absent throughout the manuscript. For example, can 2 minutes of a self-selected walking pace stress sufficiently the oxygen transport physiology? What are your thoughts considering responses to (aerobic) exercise/activity? Otherwise, it only may look that you are “fishing” correlations. 

Specific comments

Line 31: “Such” is missing between “parameters” and “as”? 

Lines 53–54: The background on this has not been sufficiently developed in the previous paragraphs for readers appraise the relevance of conducting this kinematics.

Line 57: Why was a sample size calculation not performed? Please, explain.

Lines 59–60: Can you briefly describe the work-up conducted by the physician? As if other researchers could replicate your procedures/study.

Line 62: “subjects” instead of “the subject”.

Lines 64–105: Please add the measurement properties/repeatability/reproducibility of the measures used in this investigation. Also, briefly explain data processing regarding sophisticated instruments (e.g., force platform, optical system, etc.)

Line 67: Was “…until exhaustion…” the sole criterion to end up the maximal incremental test? Please, explain.

Lines 70–72: Is this (blood samples) aligned with your study goals (lines 50–54)? Please, explain.

Line 73: Please, state the model and manufacturer of both instruments. Body weight would be more proper to state than body mass.

Line 84: This needs the reference otherwise you must write the procedures.

Lines 85–86: What R stands for in ROE and RCE?

Lines 108: Explain why Pearson’s r (a parametric test; more robust) was not used instead.

Table 2: You have repeated information (2MWT, 5TSST, handgrip strength). Please check “Roemberg”. I believe it’s misspelled.

Line 136: What do you mean by “mechanical”? Gait mechanics/kinematics?

Line 143: Why are you stressing metabolic disorders if previously you reported that “…a physician conducted an anamnesis to exclude those who presented any clinical condition…” (lines 59–60)?

Lines 140–142: Well, this is only math. Math only supports the decision, doesn’t construct the rationale. What are the potential physiological mechanisms (ir)responsible for such findings? In other words, why haven’t you found a strong relationship? Please, discuss.

Based on your research, it looks like the 2MWT is somewhat redundant (considering other physical performance/capacity measures) not offering, apparently, any superior or complementary assistance to clinical decision. I seems that the only advantage may be the (lesser) length of the corridor needed to perform it but it does not offer a similar information as the 6MWT, is more time consuming than the 5TSST and the potential use of the kinematics for gait assessment is likely of marginal use in clinical practice as a kinematics system is not available in most settings. I’m not seeing the usefulness of this test. Can you explain?

Comments on the Quality of English Language

I provide some examples of missing words or misspelling in my "Comments and Suggestions for Authors
"

Author Response

Please, see the attached document. 

Reviewer 2 Report

Comments and Suggestions for Authors

Overview 

The authors conducted a cross-sectional study to determine whether a 2-min walking test for maximum distance correlated with cardiovascular fitness as quantified with maximum oxygen consumption.  They tested 17 older (mean age 76 y) healthy individuals for distance in a 2-min walk test, stride length and gait speed, VO2 max on a progressive workload ergometer test, balance tests, and a sit-to-stand for 5 repetitions, and maximum handgrip test. The authors did not find a statistically significant relationship between distance walked in 2 min and the maximum oxygen uptake during the ergometry test.  Leg power at maximum VO2 also was not correlated with the 2-min walking distance.  As expected, stride length and step frequency were correlated with distance walked.  Hand grip was not correlated but the time to sit-and-stand five times was correlated with the 2-min walk results.  The authors concluded that the 2-min walk is not an index of aerobic fitness and may be related to leg power.

Major Concerns 

The authors state in the abstract (lines 11-13) and introduction (lines 46-47) that no prior studies have been done on the correlation between the 2-min walk and aerobic fitness.  This reviewer found 3 studies with a brief search on PubMed, the first and second below being quite relevant:

·        Association between maximal oxygen consumption and physical performance tests among older adults with cognitive frailty. Ibrahim A, Mat Ludin AF, Shahar S, Hamzah NH, Chin AV, Singh DKA.Exp Gerontol. 2023 Dec;184:112326. doi: 10.1016/j.exger.2023.112326. Epub 2023 Nov 18.PMID: 37967590 

·        The 2-minute walk test is not a valid method to determine aerobic capacity in persons with Multiple Sclerosis. Beckerman H, Heine M, van den Akker LE, de Groot V.NeuroRehabilitation. 2019;45(2):239-245. doi: 10.3233/NRE-192792.

·        Somewhat related: Estimating cardiorespiratory fitness in well-functioning older adults: treadmill validation of the long distance corridor walk. Simonsick EM, Fan E, Fleg JL.J Am Geriatr Soc. 2006 Jan;54(1):127-32. doi: 10.1111/j.1532-5415.2005.00530.x.PMID: 16420209

In the Introduction, the authors imply twice that the 6-min walk (lines 48-49 and  50-52) would also be compared to the 2-min walk and VO2 max testing.  At least, that is the interpretation by this reviewer.  Please revise these sections so as not to mislead readers.

The criterion outcome, maximum oxygen consumption, was assessed using a cycle ergometer.  While this is an effective way to quantify leg power during a fitness during a progressive workload test, is cycling a relevant standard for evaluating a walking test?  Different leg muscles are used, and this could be partially why no correlation was detected.  This should be addressed in the Discussion.

Specific Comments/Suggestions

Line 17: The authors state testing was done “over 3 different days.”  However, line 65 of the Methods states “(tests) conducted within a time span of 2 weeks.”  Which is correct?  Please clarify and make consistent.

Line 21: The authors state that the 2-min walk is associated with lower limb power but Table 2 indicates the correlation between leg power (90 w average) and distance for the 2-min walk was not statistically significant.  Please revise or explain how the authors’ conclusion is justified.

Line 57 “A convenience sample of 17 older adults…”  What was the breakdown for males and females?  I didn’t see it mentioned in the manuscript. 

Lines 81-82 “the oxygen consumption at ventilatory thresholds 1 (VT1, 82 mL·kg-1·min-1) and 2 (VT2, mL·kg-1·min-1),” Please add the definitions of VT1 and VT2.  Also, what criteria was used to evaluate whether the participants achieved a true maximum oxygen consumption during the cycling test?  What were the RPE, heart rate, and the respiratory exchange ratio at max?  Was encouragement given during the test?  What defined fatigue or failure to continue at max?

Lines 94-98 “subjects performed 2 repetitions of the 5 times sit-to-stand…:” Was the average time or the best (shortest) time reported for results and used for correlations?

Table 2 results, Lines 130-131: 

·        Several rows of Table 2 are duplicated.

·        Consider presenting VT as a percentage of VO2 max, in the table or in the discussion.

·        Add the definition of VT1 and VT2 to the footnotes.

Line 139 “This is contrary…”  The findings are not contrary since the authors used a different test, not the 6-min test.

Lines 144-146 “the performance in the 2-MWT exhibited moderate correlations with the stride length during the test and the 5-TSTS test, therefore suggesting that performance in this test is associated with lower limbs’ power.”  The authors need to provide references supporting that stride length is associated with leg power.  The authors direct data (leg power presented in table 2) is not associated with the 2-min walk distance.  I see in the 1st sentence of the next paragraph reference 9 is provided.  However, the authors need to discussion the discrepant results with the direct measure of power on the ergometer.

Line 154-155 “…the performance in the 2-MWT exhibited moderate correlations with the stride length during the test and the 5-TSTS test, therefore suggesting that performance in this test is associated with lower limbs’ power.” I don’t agree with this statement based on maximum power reported in Table 2.

End of Discussion: Limitations of the study need to be addressed and discussed.  For example:

·        Ergometry, not treadmill walking, was used for the criterion outcome of comparison for comparison with a walking activity.

·        Unless the authors can provide data, lack of measures independent of VO2 max values, i.e., RPE at max, max HR, and RER at max, leave the readers with lack of confidence that true VO2 max was achieved in all subjects.

·        Males and females were combined (if true).  This is not necessarily a major problem but with the relatively small sample size, the correlations could be inflated.  For this reason, it might be of value to provide a scatterplot of 2-min walk distance vs VO2 max and 2-min walk distance vs time for 5 sit-and-stands.

Comments on the Quality of English Language

Assistance is needed in a n umber of sections of the paper.

Author Response

Please, see the attached document. 

Round 2

Reviewer 1 Report

Comments and Suggestions for Authors

Authors have performed substantial improvements in the reporting, addressing (almost) all my major concerns. I insist that blood samples are not well contextualised for many potential readers (e.g., bachelor students in the field of Sports and Health Sciences) and not aligned with authors’ (just this) study objectives (lines 94–99). If it was used as an eligibility criterion by the physician and/or authors to attest the healthy state of the participants, it should be clearly highlighted that in the text; and maybe the table of blood sample findings would better placed in the Methods section, subsection Subjects (nearby the description of the physician work-up), or, in alternative, as supplementary material, than in the section Results. I would leave the section Results just for data related with your study questions and objectives.

A detail for consideration: authors use spatiotemporal, spatial-temporal and spatial temporal throughout the text. Perhaps just using one of the expressions throughout would be better for readers' understanding not well familiarised with the subject.

Author Response

Authors have performed substantial improvements in the reporting, addressing (almost) all my major concerns. I insist that blood samples are not well contextualised for many potential readers (e.g., bachelor students in the field of Sports and Health Sciences) and not aligned with authors’ (just this) study objectives (lines 94–99). If it was used as an eligibility criterion by the physician and/or authors to attest the healthy state of the participants, it should be clearly highlighted that in the text; and maybe the table of blood sample findings would better placed in the Methods section, subsection Subjects (nearby the description of the physician work-up), or, in alternative, as supplementary material, than in the section Results. I would leave the section Results just for data related with your study questions and objectives.

  • We would like to thank the reviewer for the positive feedback for the revision of our manuscript. In relation to table 1, following the recommendation of the reviewer, we have decided to move it into the methods section, as part of the description of the sample. Please, note that we have also performed other few minor edits in different sections of the manuscript (highlighted in red).

A detail for consideration: authors use spatiotemporal, spatial-temporal and spatial temporal throughout the text. Perhaps just using one of the expressions throughout would be better for readers' understanding not well familiarised with the subject.

  • Thank you for this suggestion. We have standardized the term throughout the whole manuscript and decided to use the term spatial-temporal.

Reviewer 2 Report

Comments and Suggestions for Authors

Major Concerns

I appreciate the authors' response to my concerns and addressing most of them with edits and clarifications.  However, I continue to have issues with how the authors portray the 5-TSTS.  Please address my comments listed here.

Specific Comments/Suggestions

Line 19-21 “No correlations were found between aerobic fitness indices and the distance covered in 2-MWT, but significant moderate correlations were found with the time to perform the 5-times sit to stand test (rho = 0.49), and with stride length (rho = 0.52)…” Please clarify which variable is significantly correlated with the sit-to stand and stride length performances, aerobic fitness or the 2-MWT.

Line 154 “…considered a good surrogate…:”  What do the authors mean by “good?”  Consider using a more objective term.  Given this study is about prediction, do the authors mean “reliable?” Or do they mean “feasible” for an older population?

Lines 217-218 “lower limbs’ muscle power:”  Instead of power, state “physical performance.”

Lines 219-220 “Of note, the power output in this test is considerably higher than the power associated to VO2max during…(ergometer)…:” Maximum aerobic power can be quantified during the VO2 max test, as the authors have done in this study.  Is power output measured during a 5-TSTS test?  I’m not familiar with how that would be done.  I encourage the authors to change “is considerably higher” to “is presumably higher.”  Most experts in this field would agree with that, but also agree power is not quantified and there are other aspects to performing 5-TSTS.

Lines 221-222 “…(i.e. lower limbs’ peak power vs. aerobic power)…”  I suggest the authors state “peak anaerobic power” in place of “lower limb peak power.”  Peak or maximum power is power (watts) but I think the authors mean is that the metabolic demands differ depending on the duration. And that influences the performance capacity.

Line 222 “…it has been found that there is…”  If the authors are referring to their own results in this report, state it as something like “…we found a moderate…”  In other words, indicate they are communicating about their findings.  Otherwise, a reference is needed.

Lines 225-228 including “multiple muscle strength:”  I think I understand what the authors are getting at.  Why not change multiple muscle strength to “the strength required by multiple muscles…” and change “(functional strength)” to “(physical performance),” which captures strength, endurance, balance, etc. all of which are needed for daily tasks.

Lines 263-266 “These results confirm the relevance of leg’s muscle power for gait performance in the elderly while would suggest the necessity of using specific aerobic fitness testing for an appropriate evaluation of this capacity in this population.”  I don’t believe that authors have results that support this statement.  Other than aerobic power, which was not correlated to the 2-min walking test results, power was not quantified.  There is more to the 5-TSTS test than just power and power is not quantified with that test.  Please revise accordingly.

Comments on the Quality of English Language

Minor editing needed.

Author Response

Major Concerns

I appreciate the authors' response to my concerns and addressing most of them with edits and clarifications.  However, I continue to have issues with how the authors portray the 5-TSTS.  Please address my comments listed here.

  • Thank you for the time devoted to revise our manuscript and for the insightful comments which have helped to improve the manuscript. Please, note that we have also performed other few minor edits in different sections of the manuscript (highlighted in red).

Specific Comments/Suggestions

Line 19-21 “No correlations were found between aerobic fitness indices and the distance covered in 2-MWT, but significant moderate correlations were found with the time to perform the 5-times sit to stand test (rho = 0.49), and with stride length (rho = 0.52)…” Please clarify which variable is significantly correlated with the sit-to stand and stride length performances, aerobic fitness or the 2-MWT.

  • Reworded following the reviewer’s suggestion. Thanks.

Line 154 “…considered a good surrogate…:”  What do the authors mean by “good?”  Consider using a more objective term.  Given this study is about prediction, do the authors mean “reliable?” Or do they mean “feasible” for an older population?

  • Thank you for highlighting this issue. We have added “following validated formulae” at the end of the sentence for clarity as the mean power in W during this test can be estimated with different formulae including body mass, height, the performance time of the test and the height of the chair used.

Lines 217-218 “lower limbs’ muscle power:”  Instead of power, state “physical performance.”

  • Sorry but we cannot agree with the reviewer. Following the recent literature, we can state confidently that the 5TST is a lower limbs’ power test.

References

Meulemans, L., Alcazar, J., Alegre, L. M., Dalle, S., Koppo, K., Seghers, J., Delecluse, C., & Van Roie, E. (2023). Sensor- and equation-based sit-to-stand power: The effect of age and functional limitations. Experimental gerontology, 179, 112255. https://doi.org/10.1016/j.exger.2023.112255

Baltasar-Fernandez, I., Alcazar, J., Losa-Reyna, J., Soto-Paniagua, H., Alegre, L. M., Takai, Y., Ruiz-Cárdenas, J. D., Signorile, J. F., Rodriguez-Mañas, L., García-García, F. J., & Ara, I. (2021). Comparison of available equations to estimate sit-to-stand muscle power and their association with gait speed and frailty in older people: Practical applications for the 5-rep sit-to-stand test. Experimental gerontology156, 111619. https://doi.org/10.1016/j.exger.2021.111619

Baltasar-Fernandez, I., Alcazar, J., Rodriguez-Lopez, C., Losa-Reyna, J., Alonso-Seco, M., Ara, I., & Alegre, L. M. (2021). Sit-to-stand muscle power test: Comparison between estimated and force plate-derived mechanical power and their association with physical function in older adults. Experimental gerontology, 145, 111213. https://doi.org/10.1016/j.exger.2020.111213

Lines 219-220 “Of note, the power output in this test is considerably higher than the power associated to VO2max during…(ergometer)…:” Maximum aerobic power can be quantified during the VO2 max test, as the authors have done in this study.  Is power output measured during a 5-TSTS test?  I’m not familiar with how that would be done.  I encourage the authors to change “is considerably higher” to “is presumably higher.”  Most experts in this field would agree with that, but also agree power is not quantified and there are other aspects to performing 5-TSTS.

  • We have now changed those sentences following the reviewer’s suggestion. Please, consider that the anaerobic power obtained during the 5-TSTS can be estimated with different formulae. However we prefer to work with the actual performance in seconds. Thanks.

Lines 221-222 “…(i.e. lower limbs’ peak power vs. aerobic power)…”  I suggest the authors state “peak anaerobic power” in place of “lower limb peak power.”  Peak or maximum power is power (watts) but I think the authors mean is that the metabolic demands differ depending on the duration. And that influences the performance capacity.

  • We have changed to anaerobic power as suggested. Thanks.

Line 222 “…it has been found that there is…”  If the authors are referring to their own results in this report, state it as something like “…we found a moderate…”  In other words, indicate they are communicating about their findings.  Otherwise, a reference is needed.

  • We have added “in the current study” at the end of the sentence to clarify it. Thanks.

Lines 225-228 including “multiple muscle strength:”  I think I understand what the authors are getting at.  Why not change multiple muscle strength to “the strength required by multiple muscles…” and change “(functional strength)” to “(physical performance),” which captures strength, endurance, balance, etc. all of which are needed for daily tasks.

  • Thank you for the suggestion. Sorry but we cannot change the definition of functional strength as suggested as this is the one used by the authors of the reference used.

Lines 263-266 “These results confirm the relevance of leg’s muscle power for gait performance in the elderly while would suggest the necessity of using specific aerobic fitness testing for an appropriate evaluation of this capacity in this population.”  I don’t believe that authors have results that support this statement.  Other than aerobic power, which was not correlated to the 2-min walking test results, power was not quantified.  There is more to the 5-TSTS test than just power and power is not quantified with that test.  Please revise accordingly.

  • The reviewer is right. We have now changed to anaerobic power as previously suggested by the reviewer. Thanks.